# Decision Fault Tree Learning and Differential Lyapunov Optimal Control for Path Tracking

**DOI:** 10.3390/e25030443

**Published:** 2023-03-02

**Authors:** S. Subash Chandra Bose, Badria Sulaiman Alfurhood, Gururaj H L, Francesco Flammini, Rajesh Natarajan, Sheela Shankarappa Jaya

**Affiliations:** 1Department of Computer Science, Islamiah College (Autonomous), Vaniyambadi 635751, India; 2Department of Computer Sciences, College of Computer and Information Sciences, Princess Nourah bint Abdulrahman University, Riyadh 11671, Saudi Arabia; 3Department of Information Technology, Manipal Institute of Technology Bengaluru, Manipal Academy of Higher Education, Manipal 576104, India; 4IDSIA USI-SUPSI, University of Applied Sciences and Arts of Southern Switzerland, 6928 Manno, Switzerland; 5Information Technology Department, University of Technology and Applied Sciences-Shinas, Al-Aqr, Shinas 324, Oman; 6Department of Electronics and Communication, SIT Siddaganga Institute of Technology, Tumkur 572103, India

**Keywords:** optimal control, differential Lyapunov, fault detection, path tracking, autonomous vehicles, machine learning, decision trees

## Abstract

This paper considers the main challenges for all components engaged in the driving task suggested by the automation of road vehicles or autonomous cars. Numerous autonomous vehicle developers often invest an important amount of time and effort in fine-tuning and measuring the route tracking to obtain reliable tracking performance over a wide range of autonomous vehicle speed and road curvature diversities. However, a number of automated vehicles were not considered for fault-tolerant trajectory tracking methods. Motivated by this, the current research study of the Differential Lyapunov Stochastic and Decision Defect Tree Learning (DLS-DFTL) method is proposed to handle fault detection and course tracking for autonomous vehicle problems. Initially, Differential Lyapunov Stochastic Optimal Control (SOC) with customizable Z-matrices is to precisely design the path tracking for a particular target vehicle while successfully managing the noise and fault issues that arise from the localization and path planning. With the autonomous vehicle’s low ceilings, a recommendation trajectory generation model is created to support such a safety justification. Then, to detect an unexpected deviation caused by a fault, a fault detection technique known as Decision Fault Tree Learning (DFTL) is built. The DLS-DFTL method can be used to find and locate problems in expansive, intricate communication networks. We conducted various tests and showed the applicability of DFTL. By offering some analysis of the experimental outcomes, the suggested method produces significant accuracy. In addition to a thorough study that compares the results to state-of-the-art techniques, simulation was also used to quantify the rate and time of defect detection. The experimental result shows that the proposed DLS-DFTL enhances the fault detection rate (38%), reduces the loss rate (14%), and has a faster fault detection time (24%) than the state of art methods.

## 1. Introduction

Automation is important growth in the automobile industry. Autonomous vehicles (AVs) with superior driver assistance systems offer important benefits to drivers, giving novel transportation use scenarios and implementations. The five basic capabilities, such as localization, perception, planning, vehicle control, and system management, are considered for AVs to drive without human participation. AVs have an electronic system that performs the driving operations. AVs have vital features for the security of current vehicles [1]. With increasing demands for secure and quick transportation services for life-saving medical devices, there is a growing emphasis on safety, reliability, sustainability, and steadfastness, which makes it an extremely analytical and active study area among control communities.

A neural estimator-based fault tolerant control approach was developed to examine the post-fault and enhance system reliability for nonlinear robotic systems. In order to preserve the system’s stability, sliding mode control was employed through post-fault dynamics. Next, a neural network was applied to regenerate the revamping fault rate and account for the impact of the fault on the functionality of the entire system. Lastly, the Lyapunov approach was to achieve the control law and neural network learning algorithms. In this way, it was promised that the neural evaluator would match the rate of change in terms of the fault and guarantee the tracking control. Since the network modeling capability, it was claimed that a smooth time function was damaged, compromising the accuracy of the fault modeling and the fault tolerance [2].

A fuzzy control with unknown timing and actuator malfunction was proposed to address noise and fault before fault detection. Via sector nonlinearity, Takagi–Sugeno fuzzy model was created to assess the precariousness due to the larger variance in vehicle mass. In addition, a strong controller technique was introduced to solve network-influenced delays and prevent packet failures. A fault-tolerant controller was also created to lessen the actuator failure on the active steering systems of vehicles. The stability was improved by the Lyapunov stability theory. Despite improvements in stability and fault tolerance for car active steering systems, time spent on fault detection was unfocused [3].

The analysis of curved path tracking utilizing the Kalman filter and trigonometric function improved tracking performance [4]. On the other hand, a linear time-varying model was created to enhance the vehicle’s stability [5]. A study of path control techniques for autonomous ground vehicles was examined [6]. Additionally, model predictive control (MPC) was introduced to precisely control vehicle constraints and address the conflict that can arise between tracking and safety measures. The transformation technique used in connected and autonomous vehicles (CAVs) has the potential to reduce accidents involving more cars, enhance the quality of life, and increase the efficiency of transportation networks as a whole [7].

The merits and downsides of the most recent developments in the field of CAVs were thoroughly examined in [8]. A few of the demands, possibilities, and long-term prospects associated with CAVs were examined in [9]. Fault incidence, however, was not covered. Multiple positioning modules for automated cars were shown in [10] via residue. Despite improvements in fault detection, there was no distinction between malfunctioning and normal situations. The linear quadratic regulator technique was used to address this problem, which in turn addressed stability under both abnormal and typical traffic patterns [11].

### 1.1. Motivation

Path tracking control enables autonomous vehicles to move in a precise and secure manner and to behave safely in all driving situations. However, the relative research on the autonomous vehicle for route tracking and fault detection is quite limited due to the strong network modeling ability and fault detection rate due to constrained localization and path planning. The goal of this study is to address the issue of fault detection and course tracking for autonomous vehicles. The aforementioned concerns prompted the current study. Motivated by the above references, we focus on the Differential Lyapunov Stochastic and Decision Fault Tree Learning approach for autonomous vehicles, taking into account positional factors, complex environmental factors, day and nighttime lighting patterns, high fault detection rates, and time constraints.

### 1.2. Contribution in Paper

The following are some of the contributions made by this paper:(1)For path tracking with accurate vehicle state using a Luenberger observer and optimal steering using a ceaseless linear model, a Differential Lyapunov Stochastic Optimal Control (SOC) with customizable Z-matrices is given.(2)To increase the fault detection rate with the least amount of time required, decision fault tree learning is employed to acquire the unexpected deviation caused by the fault.(3)Simulations and field tests are used to validate the proposed DLS-DFTL method and the Decision Fault Tree Autonomous Vehicle Fault Detection algorithm.

### 1.3. Structure of the Paper

In conclusion, this study establishes an unanticipated deviation caused by fault while simultaneously analyzing the optimal control and fault detection of the vehicle. This paper is organized as follows: The related works in the areas of autonomous vehicle path tracking and fault detection are presented in Section 2. The Differential Lyapunov Stochastic and Decision Fault Tree Learning (DLS-DFTL) method’s design is discussed in Section 3. The performance measure is covered in detail in Section 4 with the help of a table and graph structure. Finally, Section 5 concludes the essay.

## 2. Related Works

The trend in autonomous vehicle technology has fast changed to more advanced strategies with the development and upswing witnessed in deep learning-based techniques. In particular, tracking and defect detection for autonomous self-driving systems have advanced quickly. A deep learning model was developed to address the issue of local position estimation in autonomous vehicles [12]. Fusion algorithms were used to study another strategy for handling faults during bad weather [13]. Linear matrix inequality criteria are used to solve problems with uncertainties and disturbances [14]. An analysis of autonomous vehicles was published [15].

In order to ensure safe driving and handling limits utilizing the G-G diagram (i.e., grip during braking and acceleration), a route-tracking controller for autonomous vehicles was created [16]. Phase Portrait was used to obtain the G-G diagram, and validation was then carried out using an FSAE racing car to guarantee the effectiveness of the suggested controller.

Additionally, compared to the standard driving model, errors are found to be much more susceptible due to the differing dynamics and characteristics of autonomous vehicles. This problem was addressed by first modifying the Stanley controller and then integrating it with an improved particle swarm model, therefore reducing the total lateral error rate [17]. In order to reduce the path tracking error, a navigation method based on the Differential Global Positioning System (DGPS) was proposed in [18]. A low-cost nonlinear cycling vehicle model was created to prevent rollover mishaps caused by trajectory tracking control models [19]. It was suggested to conduct a broad inquiry for an autonomous navigation system using deterministic and non-deterministic algorithms [20]. Finally, an external navigation system was combined with an indirect Kalman filter during field road drives in order to accomplish effective automated driving [21]. Novel fault-tolerant, finite-time, and chatter-free approaches were investigated to enhance convergence and stability. However, the fault detection rate was not measured [22]. A novel active fault tolerant control scheme was developed for handling component/actuator faults [1]. A new adaptive event-triggered mechanism was discussed for obtaining a suitable compromise between control performance and real-time transmission consumption [23]. The Lyapunov stability theory was used to minimize the loss rate. However, the fault detection time was enhanced [24,25].

Motivated by the aforementioned research, the Differential Lyapunov Stochastic and Decision Defect Tree Learning (DLS-DFTL) method was used in this paper to construct an effective path-tracking and fault-detection model for autonomous vehicles.

## 3. Using Decision Fault Tree Learning and Differential Lyapunov Stochastic Analysis

For its safe and reliable operation, a system that is expressly regarded as safety-critical depends heavily on fast processing equipment with complex control mechanisms. If an early fault in this system remains unfixed, it could lead to system collapse, which could cause harm to people, economic hardship, or even an impact on the entire environment.

In order to create a path-tracking model for a given target vehicle, a Differential Lyapunov Stochastic Optimal Control (SOC) with movable Z-matrices is provided in this study. Additionally, the method was created in a way that deals with noise and addresses the problems with faults that arise from localization and path planning. In order to support the safety arguments with realistic autonomous vehicle ceilings, a suggested trajectory model is presented for this purpose. The Decision Fault Tree Learning (DFTL) fault detection algorithm was then provided to find an unexpected deviation caused by a fault.

The suggested Differential Lyapunov Stochastic and Decision Fault Tree Learning (DLS-DFTL) approach for rapid and precise fault identification is shown schematically in Figure 1.

The path tracking for autonomous vehicles is divided into two sections, as shown in the above diagram: localization, Differential Lyapunov Stochastic Optimal Control (SOC) path planning with configurable Z-matrices, and Decision Fault Tree Learning fault detection (DFTL). The following sections contain a detailed discussion of the two models mentioned above.

### 3.1. Adjustable Z-Matrices in Differential Lyapunov Stochastic Optimal Control (SOC)

The figure depicts the suggested Differential Lyapunov Stochastic Optimal Control (SOC) for path tracking with movable Z-matrices. By using the Luenberger observer, the precise status of the vehicle is assessed after acquiring the path information. The best steering angle is then determined using a perpetual linear model. A precise, powerful vehicle model is necessary to enhance the performance of the Luenberger observer and the unceasing linear model.

With the aid of an accurate power vehicle model, the Luenberger observer is plotted while correlating it with sensor noise, and a linear model with a movable Z-matrix is created. Based on dynamic elements related to the vehicle’s speed, a customizable Z-matrix is shown. In this way, the noise, accuracy, and fault tolerance situations in autonomous vehicle localization and path planning are manageable. Figure 2 depicts the Differential Lyapunov SOC with reversible Z-matrices in schematic form.

The function used to steer a road vehicle is designed to give a ceaseless linear model the best possible control, and it is formulated as follows.
(1)a=Pa+QIn+Rj
(2)b=Sa+Ti+Uj

In Equations (1) and (2) above, “*a*” is the array of “*n*” state variables, “*In*” is the the control input, “*b*” is the control output, and “*P*” through “*Q*” are the matrices with infinitely many coefficients. Here, the value of “*In*” is determinedn to forecast output to “*b*(*t*)” in order to reach a target “*b* des (*t*)” during a specific time period; “*t*” remains the control objective. Preliminary conditions are “*a* 0” at time “*t* = 0”, and an input I and a significance noise factor “*j*” are required for the endless linear model. The time response is then mathematically stated as follows.
(3)$$a\left(t\right)=ST_{t}a_{0}+\int_{0}^{t}ST_{\left[m*m\right]}QIndm+\int_{0}^{t}ST_{\left[m*m\right]}Rjdm$$

According to Equation (3) above, “*ST*” is the state transition matrix at time “*t*”, and “*ST_t_*” is the state transition matrix at time “*m*m*”. Each coefficient in the state transition matrix represents a portion of the state variable “*x*” at time “*t*” that is linearly connected to the state variable “*y*” at time zero and was differentiated with “*dm*”. By combining Equations (2) and (3) above, the output response is as follows:(4)$$b\left(t\right)=S*ST_{t}a_{0}+S\left[\int_{0}^{t}ST_{\left[m*m\right]}\right]\left[QIn+Rj\right]$$

Next, a Luenberger observer was created, which estimates the states of an observable system while taking error and noise into account during localization and planning (i.e., autonomous vehicle path tracking). The aforementioned endless linear model and the Luenberger observer use discrete input, discrete output, and discrete state space isolate and address the problems with error and noise that impact localization and planning. Additionally, because the suggested method is a dynamic model, it is prospective even with changes in vehicle requirements that have been seen.

In this way, the tracking issue and the noise that are present during localization are also effectively resolved. This is mathematically expressed as follows:(5)$$SV^{\prime }\left(k+1\right)=ST_{d}SV^{\prime }\left(k\right)+In_{d}In^{\prime }\left(k\right)+O_{d}\left[b\left(k\right)-b^{\prime }\left(k\right)\right]$$

According to Equation (5) above, “*SV*” (*k*) refers to the “*k*” evaluated state vector for the “*k*th” input vector “*In*” “(*k*)”, and the “kth” evaluated output vector “*b*” “(*k*)” through discrete state matrix “*ST_d_*”; discrete input matrix “*In_d_*”; and discrete observer matrix “*O_d_*”, respectively. Last but not least, the error rate is mathematically represented as follows:(6)$$e\left(k+1\right)=\left(ST_{d}-O_{d}OM_{d}\right)*e\left(k\right)$$

According to Equation (6) above, the term “*e*(*k*)” is the “*k*th” error vector for the appropriate output matrix “*OM_d_*”. The error, in this case, coincides with zero when “ST-O-OM” has eigenvalues that are closer to “1”. After examining the tracking characteristics, a configurable *Z*-matrix was used to determine the relevant distance. The changeable *Z*-matrix, which may be mathematically stated as follows, gives a tracking property for various vehicle speeds and offers a modest ceiling for the autonomous vehicle without the need for tuning.
(7)$$Z=OM=\left(om_{ij}\right),\;where\;om_{ij}\leq 0\;$$

The last step was to use a Lyapunov function as a trajectory-tracking function that produces a steady advancement in relation to an equilibrium point. The trajectory function is a Lyapunov with movable *Z* matrices for a specific target vehicle according to the suggestion trajectory generation model. It is possible to optimize the trajectory by choosing Lyapunov with changeable *Z*-matrices appropriately. The following is how this is expressed mathematically:(8)$$L^{\prime }\left(SV_{n+1}\right)=\min L\left[f\left(SV_{n},a_{n}\right)\right]$$

The best action “*a_n_*” from the starting state (i.e., source state) to the equilibrium point was chosen using the function “*L*” () and state vector “*SV_n_*” from Equation (8) above. The following is a list of the Differential Lyapunov Stochastic Optimal Control Localization and Path Tracking’s pseudo code representations as shown in Algorithm 1.
**Algorithm 1.** Path tracking and Differential Lyapunov Stochastic Optimal Control Localization**Input**: control input “In=VID,VPosx,VPosy,Temp,FSpray”,**Output**: Dependable and precise route trackingStep 1: **Initialize** ceaseless coefficients “P, Q, R, S, T, U, V”;Step 2: **Begin;**Step 3: **For** each control input “i”;Step 4: Evaluate array of “n” state variable using Equation (1);Step 5: Evaluate control output using Equation (2);Step 6: Evaluate ceaseless linear time response using Equation (3);Step 7: Evaluate output response using Equation (4);Step 8: Evaluate state vector for tracking using Equation (5);Step 9: Obtain the error rate using Equation (6);Step 10: Produce adjustable Z-matrix using Equation (7);Step 11: Evaluate Lyapunov function for path tracking using Equation (8);Step 12: **End for;**Step 13: **End.**

Two different phases are necessary to provide reliable and accurate path tracking, as stated in the method above. The algorithm takes into account the vehicle ID, the vehicle’s horizontal position, and the vehicle’s vertical position, using a never-ending linear model and tracking dynamics with changeable Z-matrices. State space and error are first seen using the Luenberger observer. Next, the error rate is reduced with the changeable Z matrix values, allowing for effective path tracking.

### 3.2. Decision Fault Tree Learning (DFTL)

Decision Fault Tree Learning is used to detect the presence of any unexpected deviation caused by a fault with the consequent optimal control localization and path tracking (DFTL). Figure 3 depicts the DFTL model’s schematic view.

The training data for autonomous path tracking and fault detection consist of “*X*, *Y*, and *W*”, as shown in the above image. Here, “*X*” stands for the payload set and is mathematically represented as follows.
(9)$$X=In=\left\{x_{1},\;x_{2},\ldots ,x_{n}\right\}$$

The payload set for each autonomous vehicle includes, according to Equation (9), the vehicle ID (VID), vehicle horizontal position (VPos *x*), vehicle vertical position (VPos *y*), sunset (SSet), and daylight (DLight), in that order. The control inputs are obtained the same regardless of the number of cars. Next, “*Y*” is the outcome value determining whether a defect was discovered or not, “Y0,1”. Finally, “*W*” is the weight of “*n*” control input training data and is shown as follows.
(10)$$W=\left\{w_{1},\;w_{2},\ldots ,w_{n}\right\}$$

The decision tree is claimed to have been created using the weighted control input values that were previously assumed. Then, it chooses a random car from among all of them. Finally, it divides the control input training data into “*V_l_*” and “*V_r_*” and assesses information gain “*IG*” for developing a decision tree, which is stated as follows.
(11)$$\Delta IG=IG\left(V_{n}\right)-\frac{V_{l}}{V_{n}}IG\left(V_{l}\right)-\frac{V_{r}}{V_{n}}IG\left(V_{r}\right)$$

In Equation (11), “*V_l_*” denotes the value of the left child in “*V_n_*”, and “*V_r_*” denotes the value of the right child in “*V_n_*”. Since the information gain value for factors such as sunset and sunshine are measured separately, the left and right children are extended accordingly. The evaluation of “*IG*(*V*)” follows, as shown below.
(12)$$IG\left(V\right)=1-\sum_{i\in V}W_{i}/\sum_{i\in V}W_{i}$$

The distance between two points (VPos *x*, VPos *y*) is measured for obtaining divergences using Bregman and is mathematically represented as follows. This is performed with the help of information gain value.
(13)$$D_{F}\left(VPos_{x},VPos_{y}\right)=F\left(VPos_{x}\right)-F\left(VPos_{y}\right)$$

The diagnostic rate for finding flaws is mathematically expressed as follows using the obtained divergences.
(14)$$dr_{t}=\sum_{y_{i}\neq y_{i}^{\prime }}W_{i}/\sum_{i=1,2,\ldots ,n}W_{i}$$

The number of vehicles taken into account for simulation in the communication network is the denominator in Equation (13), and the numerator is determined depending on the performance of the vehicle in the communication network. The numerator is the number of defective vehicles in the communication network if the result of the network is “GOOD”. The number of regular cars in the communication network is represented by the numerator if the results of the communication network are “BAD”.

The following provides the Decision Defect Tree Learning (DFTL) pseudo code representation for early autonomous vehicle fault detection as depicted in Algorithm 2.
**Algorithm 2.** Autonomous Vehicle Fault Detection Using a Decision Fault Tree**Input**: Input “In=VID,VPosx,VPosy,SSet,DLigh”**Output**: Early fault detection Step 1: **Begin;**Step 2: **For** each Input “In”;Step 3: Acquire payload data and weight using Equations (9) and (10);Step 4: Measure information gain for each vehicle using Equation (11);Step 5: Measure distance between two consecutive points (i.e., VPosx,VPosy) using Equation (13);Step 6: Evaluate diagnostic rate for identifying faults using Equation (14);Step 7: **Return** (number of faulty vehicles “fv”, number of normal vehicles “nv”);Step 8: **End for;**Step 9: **End.**

As stated in the aforementioned Decision Fault Tree Autonomous Vehicle Fault Detection algorithm, the goal remains in detecting an unexpected deviation caused by fault using a Bregman Divergent Decision Tree and control input (i.e., vehicle ID, vehicle horizontal position, vehicle vertical position, sunset, and daylight). Prior to creating a decision tree based on the control input values, information gain is first measured. The distance between each vehicle is then determined for two distinct coordinates, i.e., vehicle horizontal position and vehicle vertical position, using Bregman Divergence. Finally, the diagnostic rate is used to classify defective and healthy automobiles.

## 4. Performance Evaluation

The performance analysis of the Differential Lyapunov Stochastic and Decision Fault Tree Learning (DLS-DFTL), as well as the currently used neural estimator-based fault-tolerant control [2] and fuzzy control uncertain time-delay active steering [3], are shown in this part. When the number of test datasets is varied, the fault detection rate, fault detection time, and test loss for various cars are measured using the DLS-DFTL and two current methods [2,3] for comparison. By using the TME Motorway Dataset and the Python programming language, the analysis is carried out [22].

The dataset used to benchmark DLS-DFTL was acquired utilizing the BRAiVE test vehicle in Northern Italy in December 2011 in collaboration with VisLab (University of Parma, Italy). The “TME Motorway Dataset” was made up of an image acquisition process that includes stereo; 20 Hz frequency2; 1024 × 768 grayscale lossless compressed images, with Bayer coded color information3 and a 32-degree horizontal field of view; an ego-motion estimate (a confidential computing method); and vehicle annotation and classification produced by a laser scanner.

The data shown here are time-stamped and consist of 28 clips that were chosen out of a total of about 27 min of acquisition. This selection procedure takes into account various traffic patterns, lane configurations, degree of road curvature, and lighting. In addition, the dataset has been divided into two separate subsets based on the type of lighting, including daylight and sunset.

### 4.1. Scenario 1: Performance Evaluation of Defect Detection Rate

The defect detection rate is the first important factor for tracking autonomous vehicles. Here, the term “fault detection rate” refers to the monitoring of an autonomous vehicle and the measurement of the fault detection rate in the event that the path being tracked has a fault. Here, the simulation’s driverless vehicle becomes transformed into frames. The following is how this is mathematically stated.
(15)$$FD_{rate}=\sum_{i=1}^{n}\frac{F_{F}}{F_{act}}*100$$

The fault detection rate, or “*FD_rate_*”, is calculated using the number of defective frames found during testing, or “*F_F_*”, and the actual number of faulty frames, or “*F_act_*”, as shown in Equation (15) above. It is measured in percentage (%) terms. The fault detection rate for three approaches, including DLS-DFTL and neural estimator-based fault tolerant control [2], is listed in Table 1 below, active steering with unpredictable time delays and fuzzy control [3].

The defect detection rate in relation to 250 different frames is shown in Figure 4 above. The horizontal axis in the figure denotes the 25 frames or less, and the vertical axis is the defect detection rate expressed as a percentage (%). The defect detection rate is not directly or inversely related to the frames taken into account for simulation, according to the figure. This is because different movies were extracted at various points in time, and when those videos were translated into frames, the problem that was found also varied. As a result, as the number of frames increases, the defect detection rate neither increases nor decreases. Let us take a hypothetical situation where “5” autonomous vehicles are used to track paths and identify faults. Each autonomous vehicle is divided into five frames, making a total of twenty-five frames available for simulation. In the aforementioned scenario, “3” autonomous vehicles were found to be defective.

The results of simulations for 25 frames indicate that while there were “5” total problematic frames observed, “4” faulty frames were discovered using DLS-DFTL, “3” faulty frames were discovered using [2], and “2” faulty frames were discovered using [3]. From these findings, it can be concluded that DLS-DFTL has a comparative advantage over [2,3] in terms of defect detection rate. Due to the use of the Differential Lyapunov Stochastic Optimal Control (SOC) with the Variable Z-Matrix model, this has occurred. This model’s application for path tracking resulted in the calculation of the ideal steering angle using a continuous linear model. Additionally, the customizable Z-matrix is built around dynamic elements that depend on the vehicle’s speed. With this, it is discovered that the fault detection rate using DLS-DFTL is significantly better than the 23% and 52% rates using [2,3].

### 4.2. Scenario 2: Analysis of Fault Detection Time Performance

The defect detection time is the second important statistic for path tracking and fault detection in autonomous vehicles. Early detection of malfunctioning autonomous vehicles is made easier by reliable tracking performance. The measurement of fault detection time is provided below.
(16)$$FD_{time}=\sum_{i=1}^{n}F_{i}*Time\;\left[FD\right]$$

According to Equation (16) above, the fault detection time (*FD_time_*) is calculated based on the frames that are taken into account during simulation (*F_i_*) and the amount of time needed to identify a malfunctioning autonomous vehicle (*FD_time_*). It is quantified in milliseconds (ms). The fault detection times for three different methods—DLS-DFTL, neural estimator-based fault tolerant control [2], and fuzzy control uncertain time-delay active steering [3]—are listed in Table 2 below.

The three methods—DLS-DFTL, neural estimator-based fault tolerant control [2], and fuzzy control uncertain time-delay active steering [3]—are shown in Figure 5 above in terms of the time it takes to detect faults. The vertical axis depicts the fault detection time, with the horizontal axis showing the frames used for path tracking and defect detection for autonomous vehicles. The fault detection time is determined to be exactly related to the simulation frames used, according to the figure. Therefore, when the number of films used for simulation increases, the frames also increase, and faults may vary depending on the path being tracked, which inevitably lengthens the time it takes to discover faults.

However, simulations with 25 frames revealed that the fault detection time for DLS-DFTL was 3.375 ms, 4.625 ms for [2], and 5.357 ms for [3]. According to the simulation results, implementing DLS-DFTL is observed to reduce the fault detection time when compared to [2,3]. The use of the Differential Lyapunov Stochastic Optimal Control Localization and Path Tracking method is what caused the improvement. The endless linear model, with which the state space and error were significantly observed, is the first model to which this technique is applied in order to track an exact path. The reported error rate was then decreased by utilizing configurable Z-matrices, allowing for considerable path tracking and reducing the fault detection time by 31% and 16%, respectively, in comparison to [2,3].

### 4.3. Scenario 3: Loss Rate of Performance

The loss rate is the final parameter used in the path tracking and defect detection of autonomous vehicles. The loss rate is the percentage of times that the path tracking and fault detection processes miss the malfunctioning autonomous vehicle. The loss rate is calculated as shown below.
(17)$$L_{r}=\sum_{i=1}^{n}\frac{F_{MO}}{F_{i}}*100$$

Based on the total number of frames taken into account for the simulation (*F_i_*) and the missed-out malfunctioning autonomous vehicle’s frame (*F_MO_*), the loss rate (*L_r_*) is calculated using Equation (17) above. It is measured in percentage (%) terms. The loss rate for each of the three methods—DLS-DFTL, neural estimator-based fault tolerant control [2], and fuzzy control uncertain time-delay active steering [3]—is listed in Table 3 below.

The loss rate for three distinct approaches is shown in Figure 6 above. The chart suggests that increasing the frames causes the loss rate to increase significantly. The loss rate in this context means that the malfunctioning autonomous vehicle went unnoticed. If there are fewer defective autonomous vehicles that go unnoticed, tracking accuracy will not be as good. Additionally, the figure shows that raising the frame count likewise raises the loss rate. However, when compared to [2,3], a comparative study carried out through simulations demonstrates better results when employing DLS-DFTL. In other words, simulations with 25 frames reveal that 85 use DLS-DFTL, while 12% and 16%, respectively, use [2,3].

Due to the implementation of the Decision Fault Tree Autonomous Vehicle Fault Detection algorithm, the loss rate is said to be decreased when utilizing DLS-DFTL. This method takes into account three different variables. First, information on the anonymous vehicle’s horizontal and vertical positions, as well as the time of day, sunset, and daylight, was used to inform the decision tree. Bregman was then used to calculate the distance between each vehicle for two distinct coordinates, i.e., vehicle horizontal position and vehicle vertical position, and finally, the diagnostic rate was used to distinguish between the faulty and normal autonomous vehicles.

## 5. Discussion

Though there are numerous talented approaches for tracking control of fault-tolerant methods in the literature, there is still room for the development of existing methods. The aforementioned concerns have prompted the current study. Furthermore, the majority of tracking control methods have important flaws that make their real-world implementation challenging. Hence, this study proposes Differential Lyapunov Stochastic and Decision Fault Tree Learning (DLS-DFTL) for accurate path tracking. Differential Lyapunov Stochastic Optimal Control Localization was utilized to minimize fault detection time via movable Z-matrix. Decision Fault Tree Learning (DFTL) was employed to identify unanticipated deviation caused by the fault. Bregman Divergence was applied to measure distance among every vehicle for two distinct coordinates. In this way, the diagnostic rate identifies faults with lower latency.

Because of their advantages, such as guaranteed stability, ease of implementation, higher fault detection rate, lower fault detection time, and loss, the current investigation was prompted by this concern. It was revealed during numerical results that the proposed control technique is suitable for tracking control systems. This study compared the proposed DLS-DFTL with the existing neural estimator-based fault-tolerant control [2] and fuzzy control uncertain time-delay active steering [3] using TME Motorway Dataset based on various parameters, such as fault detection rate, fault detection time, and loss rate. The current results confirm that the proposed DLS-DFTL improves the fault detection rate by 38%, reduces the fault detection time by 24%, and minimizes the loss rate by 14% compared to the existing ones, namely neural estimator-based fault-tolerant control [2] and fuzzy control uncertain time-delay active steering [3] with the aid of TME Motorway Dataset.

## 6. Conclusions

In this paper, we proposed a powerful Differential Lyapunov Stochastic and Decision Fault Tree Learning (DLS-DFTL) paradigm for tracking vehicle dynamics and handling in the presence of faults. The proposed DLS-DFTL method was designed based on the Differential Lyapunov Stochastic Optimal Control (SOC) with a Variable Z-Matrix model and Decision Fault Tree Autonomous Vehicle Fault Detection algorithm. At first, the Luenberger observer combined the best features of the continuous linear model. Differential Lyapunov Stochastic Optimal Control Localization and Path Tracking were implemented for the autonomous vehicle to achieve reliable performance in the presence of disturbances. Afterward, Decision Fault Tree Learning (DFTL) was applied for early problem identification in autonomous vehicles. The DLS-DFTL approach was applied in Python for precise validation. The numerical results showed that the DLS-DFTL approach outperformed with full proof trajectory tracking, which is capable of monitoring effectively in the literature. Due to the lower loss rate and fault detection time, the DLS-DFTL approach achieved better performance in terms of fault detection rate, fault detection time, and loss rate. Finally, we also showed that the average loss of the proposed mechanisms was very small. In the future, the proposed scheme with two fault-tolerant mechanisms is a promising solution for emerging traffic management of high reliability and low latency.

## Figures and Tables

**Figure 1 entropy-25-00443-f001:**
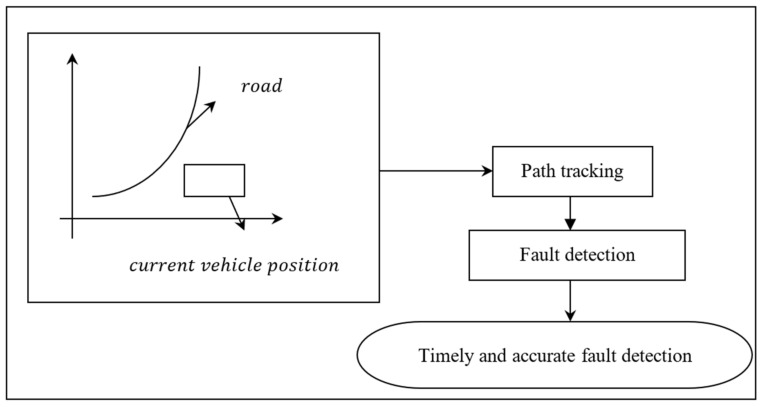
Diagram of the DLS-DFTL approach (Differential Lyapunov Stochastic and Decision Fault Tree Learning).

**Figure 2 entropy-25-00443-f002:**
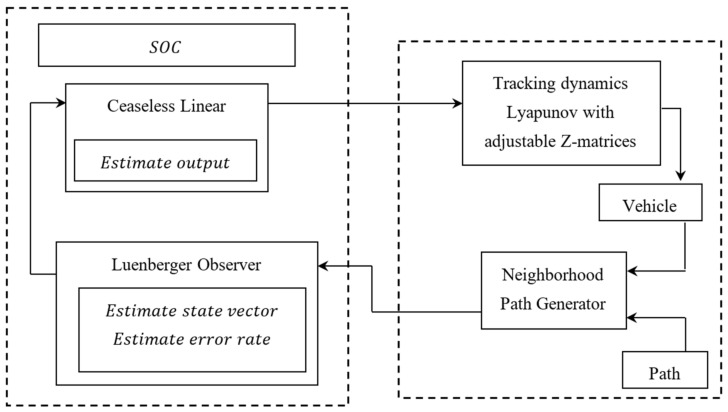
Differential Lyapunov SOC schematic with customizable Z-matrices.

**Figure 3 entropy-25-00443-f003:**
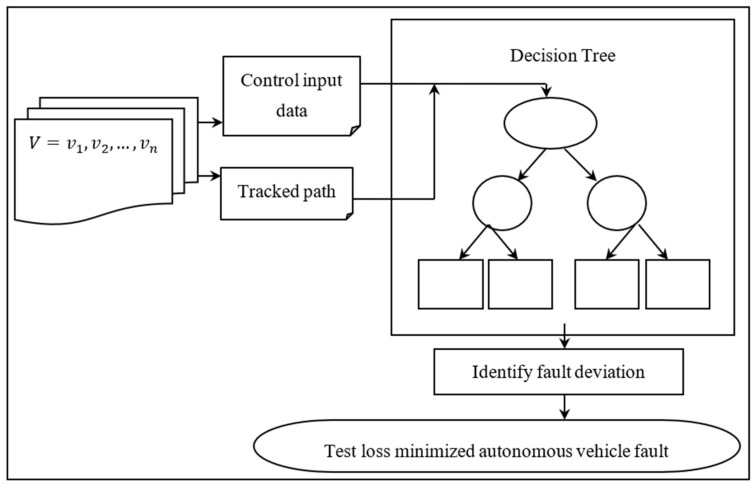
Representational view of DFTL model.

**Figure 4 entropy-25-00443-f004:**
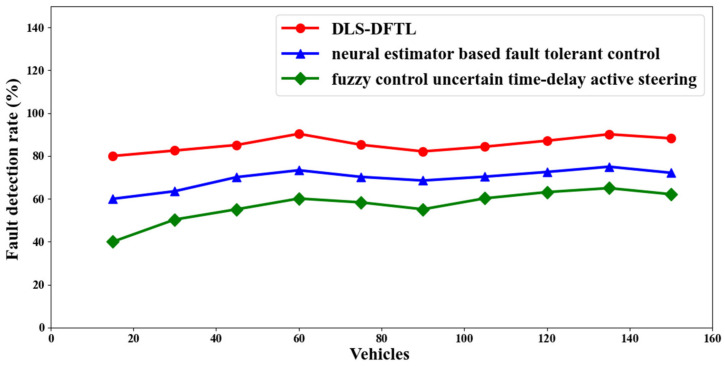
Visualization of the fault detection rate.

**Figure 5 entropy-25-00443-f005:**
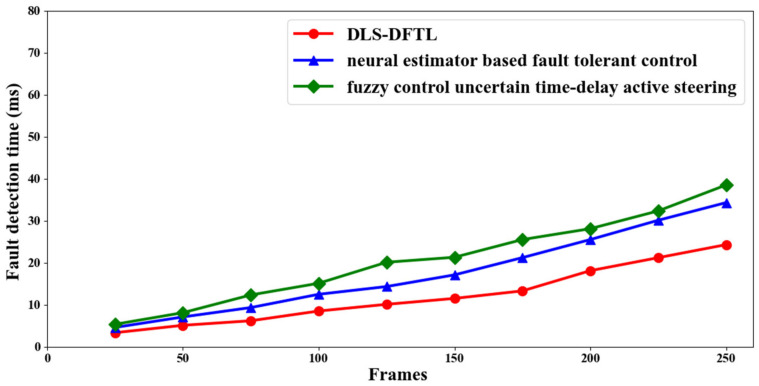
Visualization of the fault detection time.

**Figure 6 entropy-25-00443-f006:**
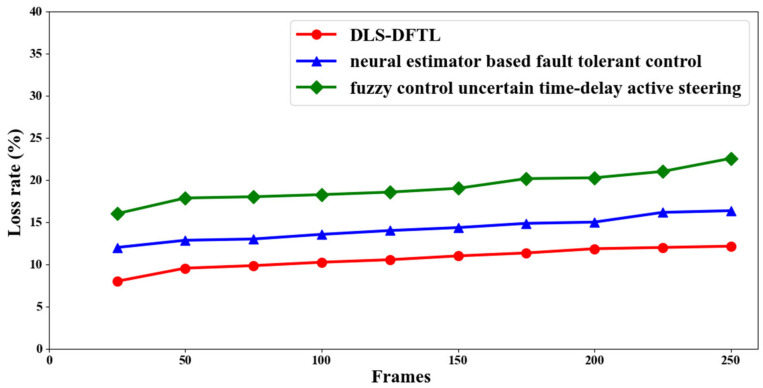
Loss rate in graphical representation rorm.

**Table 1 entropy-25-00443-t001:** Comparison of the rates of fault detection using various techniques.

Vehicles	Fault Detection Rate (%)		
	DLS-DFTL	Neural Estimator-Based Fault Tolerant Control	Fuzzy Control Uncertain Time-Delay Active Steering
15	80	60	40
30	82.55	63.55	50.35
45	85.15	70.15	55.15
60	90.35	73.35	60.15
75	85.25	70.25	58.35
90	82.15	68.55	55.15
105	84.35	70.35	60.25
120	87.15	72.55	63.15
135	90.15	75	65
150	88.25	72.15	62.15

**Table 2 entropy-25-00443-t002:** Fault detection time comparison for several techniques.

Frames	Fault Detection Time (%)		
	DLS-DFTL	Neural Estimator-Based Fault Tolerant Control	Fuzzy Control Uncertain Time-Delay Active Steering
25	3.375	4.625	5.375
50	5.135	7.125	8.135
75	6.215	9.355	12.355
100	8.535	12.515	15.135
125	10.125	14.355	20.135
150	11.535	17.135	21.325
175	13.325	21.235	25.535
200	18.135	25.535	28.125
225	21.225	30.125	32.355
250	24.325	34.325	38.525

**Table 3 entropy-25-00443-t003:** Loss rate comparison between different methods.

Frames	Loss Rate (%)		
	DLS-DFTL	Neural Estimator-Based Fault Tolerant Control	Fuzzy Control Uncertain Time-Delay Active Steering
25	8	12	16
50	9.55	12.85	17.85
75	9.85	13	18
100	10.25	13.55	18.25
125	10.55	14	18.55
150	11	14.35	19
175	11.35	14.85	20.15
200	11.85	15	20.25
225	12	16.15	21
250	12.15	16.35	22.55

## Data Availability

Not applicable.

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
