# Peer review of "Decision Fault Tree Learning and Differential Lyapunov Optimal Control for Path Tracking"

_entropy, 2023, doi:10.3390/e25030443_

Round 1

Reviewer 1 Report

This paper is well organized and written. I only have one minor suggestion. The introduction should highlight the motivation of using Differential Lyapunov Stochastic and Decision Fault Tree Learning. What problems exist in the existing methods need to be explained more clearly.

Author Response

Reviewer-1 #Correction is made in green color

This paper is well organized and written. I only have one minor suggestion. The introduction should highlight the motivation of using Differential Lyapunov Stochastic and Decision Fault Tree Learning. What problems exist in the existing methods need to be explained more clearly.

Response: [motivation is added in section 1.1]

  • Motivation

Path tracking control enables autonomous vehicles to move in a precise and secure manner and to behave safely in all driving situations. However, the relative research on the autonomous vehicle for route tracking and fault detection is quite limited due to the strong network modeling ability and fault detection rate due to constrained localization and path planning. The goal of this study is to address the issue of fault detection and course tracking for autonomous vehicles. The aforementioned concerns have prompted the current study. Motivated by the above references, we focus on Differential Lyapunov Stochastic and Decision Fault Tree Learning approach for autonomous vehicles, taking into account positional factors, complex environmental factors, day and nighttime lighting patterns, high fault detection rates, and time constraints. 

Reviewer 2 Report

Recommending for a major review. 

Author Response

Reviewer-2 Correction is made in red color

The paper is recommended for the major review:

Major comments:

  1. The abstract should be revised and make the problem statement more clearly.

Response: [Correction is made in abstract]

Abstract: This paper considers the main challenges for all components engaged in the driving task suggested by the automation of road vehicles or autonomous cars. Numerous autonomous vehicle developers often invest an important amount of time and effort in fine-tuning and measuring the route tracking to obtain reliable tracking performance over a wide range of autonomous vehicle speed and road curvature diversities. However, a number of automated vehicles were not considered for fault tolerant trajectory tracking methods. Motivated by this, the current research study of the Differential Lyapunov Stochastic and Decision Defect Tree Learning (DLS-DFTL) method is proposed to handle fault detection and course tracking for autonomous vehicles problems. Initially, Differential Lyapunov Stochastic Optimal Control (SOC) with customizable Z-matrices is to precisely design the path tracking for a particular target vehicle while successfully managing the noise and fault issues that arise from the localization and path planning. With the autonomous vehicle's low ceilings, a recommendation trajectory generation model is created to support such a safety justification. Then, to detect an unexpected deviation caused by a fault, a fault detection technique known as Decision Fault Tree Learning (DFTL) is built. The DLS-DFTL method can be used to find and locate problems in expansive, intricate communication networks. We have conducted various tests and showed the applicability of DFTL. By offering some analysis of the experimental outcomes, the suggested method produces significant accuracy. In addition to a thorough study that compares the results to state-of-the-art techniques, simulation has also been used to quantify the rate and time of defect detection. The experimental result shows that the proposed DLS-DFTL enhances the fault detection rate (38%), reduces the loss rate (14%) and faster fault detection time (24%) than the state of art methods.

  1. The introduction should be revised thoroughly.

Response:

Correction is made in introduction.

  1. The author is recommended to refer to the “Sensor-Based Prognostic Health Management of Advanced Driver Assistance System for Autonomous Vehicles: A Recent Survey” in the introduction of the manuscript.

Response:  [above reference is added in introduction]

Automation is important growth in the automobile industry. Autonomous vehicles (AVs) with superior driver assistance systems offer important benefits to drivers, as too giving novel transportation use scenarios as well as implementations. The five basic capabilities such as localization, perception, planning, vehicle control, as well as system management considered for AVs to drive without human participation. AVs have an electronic system that performs the driving operations. AVs have vital features for the security of current vehicles [22]. With increasing demands for secure and quick transportation services for life-saving medical devices, there is a growing emphasis on safety, reliability, sustainability, and steadfastness, which makes it an extremely analytical and active study area among control communities.

  1. The figures are not clear it should be redrawn as figure 1, 2,3 ……. and so on?

Response: 

Correction is made in all figures.

  1. The importance of the results should be clarify in terms of figures?

Response: 

Yes, the visual comparison of results is provided in terms of figures.

  1. The results should be revised?

Response: 

Correction is made in results.

  1. How the frames are categorized with 25 frames gape?

Response: 

For experimental purpose, the framed are categorized with 25. So, the proposed DLS-DFTL approach is for monitoring effectively with full proof trajectory tracking and ensuring fault detection rate and time with a low loss rate.

  1. The conclusion of the paper should rewrite, it is not that effective in the current form?

Response:  [Correction is made in conclusion]

  1. Conclusions

In this paper, we proposed a powerful Differential Lyapunov Stochastic and Decision Fault Tree Learning (DLS-DFTL) paradigm for tracking vehicle dynamics and handling in the presence of faults. The proposed DLS-DFTL method is designed based on the Differential Lyapunov Stochastic Optimal Control (SOC) with a Variable Z-Matrix model and Decision Fault Tree Autonomous Vehicle Fault Detection algorithm. At first, the Luenberger observer combines the best features of the continuous linear model. Differential Lyapunov Stochastic Optimal Control Localization and Path Tracking are implemented for the autonomous vehicle to achieve reliable performance in the presence of disturbances. After, Decision Fault Tree Learning (DFTL) is applied for early problem identification in autonomous vehicles. The DLS-DFTL approach was applied in Python for precise validation. The numerical results showed that the DLS-DFTL approach outperformed with full proof trajectory tracking, which is capable of monitoring effectively in the literature. Thanks to the lower loss rate and fault detection time, the DLS-DFTL approach achieved better performance in terms of fault detection rate, fault detection time, and loss rate.

  1. The author should include a paragraph in the results show the importance of the current results and also it is recommended to include the challenges and future prospects of the current research work?

Response:  [Correction is made in discussion and conclusion]

  1. Discussions

Though in the literature there are numerous talented approaches for tracking control of fault-tolerant methods, there is still room for the development of existing methods. The aforementioned concerns have prompted the current study. Furthermore, the majority of tracking control methods has important flaws that done their real-world implementation challenging. Hence, this study proposes Differential Lyapunov Stochastic and Decision Fault Tree Learning (DLS-DFTL) for accurate path tracking. Differential Lyapunov Stochastic Optimal Control Localization is utilized to minimize fault detection time via movable Z-matrix. Decision Fault Tree Learning (DFTL) is employed for identifying unanticipated deviation caused by the fault. Bregman Divergence is applied for measuring distance among every vehicle for two distinct coordinates. In this way, the diagnostic rate is for identifying faults with lower latency.

Because of their advantages such as guaranteed stability, ease of implementation, higher fault detection rate, lower fault detection time, and loss. The current investigation was prompted by this concern. It is revealed during numerical results. The proposed control technique is suitable for tracking control systems. This study compares the proposed DLS-DFTL with the existing neural estimator-based fault-tolerant control [1] and fuzzy control uncertain time-delay active steering [2] using TME Motorway Dataset dataset based on various parameters, such as fault detection rate, fault detection time, and loss rate. The current results confirm that the proposed DLS-DFTL improves the fault detection rate by 38%, reduces the fault detection time by 24%, and minimizes the loss rate by 14% compared to the existing, namely neural estimator-based fault-tolerant control [1] and fuzzy control uncertain time-delay active steering [2] with aid of TME Motorway Dataset.

  1. Conclusions

Finally, we also showed that the average loss of the proposed mechanisms was very small. In the future, the proposed scheme with two fault-tolerant mechanisms is a promising solution for emerging traffic management of high reliability and low latency.

Round 2

Reviewer 2 Report

accepted